# Anti-Noise 3D Object Detection of Multimodal Feature Attention Fusion Based on PV-RCNN

**DOI:** 10.3390/s23010233

**Published:** 2022-12-26

**Authors:** Yuan Zhu, Ruidong Xu, Hao An, Chongben Tao, Ke Lu

**Affiliations:** 1School of Automotive Studies, Tongji University, Shanghai 201800, China; 2Suzhou Automotive Research Institute, Tsinghua University, Suzhou 215200, China

**Keywords:** 3D object detection, multimodal feature, anti-noise, attention mechanism

## Abstract

3D object detection methods based on camera and LiDAR fusion are susceptible to environmental noise. Due to the mismatch of physical characteristics of the two sensors, the feature vectors encoded by the feature layer are in different feature spaces. This leads to the problem of feature information deviation, which has an impact on detection performance. To address this problem, a point-guided feature abstract method is presented to fuse the camera and LiDAR at first. The extracted image features and point cloud features are aggregated to keypoints for enhancing information redundancy. Second, the proposed multimodal feature attention (MFA) mechanism is used to achieve adaptive fusion of point cloud features and image features with information from multiple feature spaces. Finally, a projection-based farthest point sampling (P-FPS) is proposed to downsample the raw point cloud, which can project more keypoints onto the close object and improve the sampling rate of the point-guided image features. The 3D bounding boxes of the object is obtained by the region of interest (ROI) pooling layer and the fully connected layer. The proposed 3D object detection algorithm is evaluated on three different datasets, and the proposed algorithm achieved better detection performance and robustness when the image and point cloud data contain rain noise. The test results on a physical test platform further validate the effectiveness of the algorithm.

## 1. Introduction

Reliable 3D object detection is the primary step to ensure the proper working of autonomous driving systems. Current autonomous driving systems commonly use camera and LiDAR for 3D object detection in order to get more accurate detection results and more applicable scenarios. The camera can output high-resolution pictures, providing rich color and texture information, but it is not easy to extract the spatial position of the object. LiDAR can provide accurate spatial point cloud information, therefore most 3D object detection algorithms [1,2,3,4,5] are based on LiDAR and achieve better detection performance. However, both sensors are susceptible to weather noise: the camera is susceptible to image texture noise and the LiDAR is susceptible to spatial point coordinate noise [6,7,8,9]. This paper focuses on rainy scenarios, as it is the most common dynamic challenging weather condition that effects vision sensors. These effects can directly blur object colors and contours in images, and produce multiple reflections to affect the observability of spatial boundaries in point clouds, which seriously degrades the performance of the object detector. Therefore, effectively fusing the information provided by both sensors is an important approach to improve the robustness of environment perception. In this paper, we propose a camera and LiDAR fusion 3D object detection algorithm based on point-guided sampling of point cloud spatial features and image semantic features, which shows significant performance improvement in rain noise datasets.

In recent years, there are some studies on 3D object detection based on camera and LiDAR fusion, which can be classified into serial type [10,11,12,13,14,15,16] and parallel type [17,18,19,20,21,22,23,24,25,26,27,28,29,30,31,32,33,34,35,36,37,38] according to the stage of fusion. The serial type method is represented by F-PointNet [13] which usually takes the image of the camera as input first and uses image object detection or semantic segmentation algorithm to get the spatial location of the object, then projects it to the LiDAR point cloud to extract the point cloud of the frustum region around the object, and finally uses the normal point cloud 3D object detection algorithm to get 3D bounding boxes. F-PointPillars [10] replaced the point cloud detection network with PointPillars on this basis and improved the detection speed. F-ConvNet [14] encoded the grouping within the frustum region and achieved better detection accuracy. This type of method can effectively reduce the number of point clouds and improve the computational efficiency, but requires high image detection performance of the front. The parallel type method, on the other hand, starts with MV3D [30], which extracts the image features of the LiDAR point cloud and camera separately, and then fuses them at different stages, and finally obtains the 3D object detection results through the fully connected layer. Another approach of parallel type methods is to introduce the correspondence between spatial points and pixel points into the feature fusion, which can improve the feature correlation. Both MVP [31] and EPNet [20] take advantage of this and achieve better detection performance. Most of the subsequent parallel type methods use this fusion method, which requires the establishment of a projection relationship between spatial and pixel points by sensor calibration, and then fusing the point or voxel features with the image features of the corresponding pixel points. However, since the point cloud is sparse while the image has higher pixel point density, it will cause the problems of resolution mismatch and uneven sampling of image features. Therefore, this paper proposes a P-FPS algorithm to project the point cloud onto the image plane and then downsample the point cloud by farthest point sampling (FPS). This method can improve the sampling rate of image features for close objects.

Attention-based methods [39] can aggregate information from different feature spaces, and DETR [40] introduced self-attention into the field of image-based 2D object detection. Some studies [41,42,43] since then have attempted to apply attention mechanisms to image-based perception, and others [44,45,46,47] have used attention mechanisms in point cloud-based applications. For image and point cloud feature fusion, the heterogeneity of the two sensors can lead to the features obtained by the feature extraction network not to be in the same feature space. In order to enable the point cloud features to fuse the most associated image features, an MFA module is proposed in this paper. This module allows point cloud features to be adaptively fused with image features from different feature spaces. 

The contributions of this paper are as follows:Aiming at the mismatch resolution between point cloud and image, a point-guided 3D object detection algorithm based on point cloud and image fusion is proposed. The algorithm is based on the point-guided strategy of PV-RCNN, which fuses the multi-stage features of two sensors and enriches the raw point cloud features. Compared with other 3D object detection algorithms, this algorithm achieves significant robustness when point clouds and images contain rain noise, and can utilize image and point cloud information more sufficiently.An MFA module is proposed for the problem of fused information interference caused by different feature spaces of images and point clouds. This module uses an attention mechanism to calculate the attention weights of multiple image features and point cloud features corresponding to each point. Based on the correlation between point cloud features and spatial features, different image features are fused adaptively.To solve the problem of insufficient image features of close objects extracted from point clouds obtained by FPS, a P-FPS algorithm is proposed. By projecting the LiDAR point cloud onto the image plane before implementing FPS, uniform sampling of the point cloud in the image plane is achieved and the image feature sampling rate of the object is improved.

The remainder of this paper is as follows: In Section 2, the related background research work is discussed. The proposed algorithmic framework and key modules are presented in detail in Section 3. Section 4 describes the experimental setup and experimental results. Finally, Section 5 presents an ablation study of the method proposed in this paper.

## 2. Related Work

**Image-driven fusion.** Currently, camera-based 2D object detection has been able to achieve an average accuracy of over 90% in publicly available datasets. However, in 3D object detection, better results can be achieved by using LiDAR. Some studies have attempted to utilize the advantage of both sensors and used a serial form of 2D detection followed by 3D detection. Qi et al. [13] proposed F-PointNets for RGB-D data, and used Fast R-CNN to generate 2D bounding boxes and delineate a frustum region in 3D space, and then performed 3D detection of point clouds in this region based on PointNet. This approach reduces the computational requirements for point cloud search on the one hand, and reduces the interference of background point clouds on detection on the other hand. However, the frustum region obtained by this method will still contain background points behind the object; Wang et al. [14] made further improvements to address this problem by using sliding regions for point cloud feature extraction separately. To solve the heterogeneity problem in camera-LiDAR fusion, some works inspired by Pseudo-LiDAR [48] fuse the pseudo point cloud generated from the camera with the point cloud acquired by LiDAR. Pseudo-LiDAR++ [49], based on Pseudo-LiDAR [48], uses the binocular camera disparity estimation principle to generate pseudo point clouds and improve the quality of pseudo point clouds by matching them with LiDAR point clouds through K-Nearest Neighbor (KNN).

**Feature fusion.** Unlike the image-driven approach, MV3D [30] and AVOD [26] use the projection map of the LiDAR point cloud and the camera image as inputs, extract image and point cloud features separately, and fuse the proposal regions to obtain the final 3D detection results, avoiding the dependence on the camera detection performance. 3D-CVF [17] first transforms image features from multiple perspective to the Bird Eye View (BEV) plane, and then selects the fusion degree of the two types features by an adaptive gate fusion network. ContFuse [50] uses continuous convolution to aggregate images and BEV features at different resolutions based on geometric position relationships. This method uses dense image features to enrich the sparse LiDAR point cloud to enhance the 3D detection performance. These methods ignore the matching between features, although they use multi-layer fusion. 

**Project-based fusion.** Fusing complementary information from point clouds and images would benefit 3D detection. However, the inherent heterogeneous nature between sensors created difficulties in information fusion, and many studies projected point clouds onto the images for data sampling. If these data are processed directly, some problems are encountered: (1) sparse point clouds do not correspond to high-resolution image features one by one, resulting in incomplete data fusion; (2) point cloud data are missing depth or height information after projection. To address these problems, many attempts have been made in recent studies. Zhu et al. [37] proposed VPFNet, a network that first generates 3D proposal regions from point clouds and divides them into multiple small grids again, with the corner points of each grid as virtual points, and next projects these points onto the image for image feature sampling and point cloud feature extraction, which reduces the computation and compensates the lack of sparse point clouds. Huang et al. [20] proposed a LI-Fusion module for fusing point cloud and image information at different scales in the feature extraction stage, and then associating the image features to the corresponding point cloud locations, while adding a consistency forcing loss for balancing the impact of classification loss and location loss on the detection results. Wen et al. [36] combined the fusion framework of MV3D [30] and AVOD [26], and proposed to project the height information of the point cloud into the image to obtain RGB-D data, and then fused it with the bird’s-eye view to construct a single-stage detector while using an adaptive attention mechanism in feature extraction. Projection-based fusion methods usually capture image features based on keypoints after FPS downsampling. Due to the affine transformation, the point cloud is no longer uniform in the image plane, and the feature-rich close objects cannot be extracted completely. Therefore, this paper proposes to project the point cloud onto the image plane followed by FPS in order to sample uniform image features. Meanwhile, unlike the concat fusion of previous methods, an MFA module is proposed to make the point cloud adaptively fuse image features with different weights.

## 3. Framework of Proposed 3D Detection Algorithm

For the above problems, this paper proposes a 3D object detection method that fuses LiDAR and camera. The method can adaptively fuse image and point cloud features based on the attention mechanism to cope with the performance degradation when the point cloud or image contain rain noise. As shown in Figure 1, the network is based on PV-RCNN [1], and image features are fused into the raw LiDAR features. When using cameras and LiDAR for 3D object detection, the difference in the spatial location of the two sensors can cause the results to be in different reference coordinate systems. To associate the point cloud and image features, a projection of the point cloud to the image plane is used to map each point to the corresponding pixel location. The projection of the point $P_{L}\left(x,y,z\right)\;$of the LiDAR to the pixel point $P_{I}\left(u,v\right)$ is expressed as:(1)$$z_{c}\left[\begin{array}{c} u\\ v\\ 1 \end{array}\right]=\left[\begin{array}{ccc} \frac{f}{dx} & 0 & u_{0}\\ 0 & \frac{f}{dy} & v_{0}\\ 0 & 0 & 1 \end{array}\right]\left[\begin{array}{cc} R & T\\ 0 & 1 \end{array}\right]\left[\begin{array}{c} x\\ y\\ z\\ 1 \end{array}\right]$$
(2)$$z_{c}\left[\begin{array}{c} u\\ v\\ 1 \end{array}\right]=\mathrm{KM}\left[\begin{array}{c} x\\ y\\ z\\ 1 \end{array}\right]$$
where f is the camera focal length, dx and dy are the length and width of a single pixel at the image plane, $u_{0}$ and $v_{0}$ are offset of the pixel coordinate system, R is rotation matrix, T is translation matrix, $z_{c}$ is the depth of the point to the image plane, K is the camera internal reference matrix, and M is the transformation matrix from the LiDAR coordinate system to the camera coordinate system.

By keypoints guidance, point cloud features and image features of different scales can be merged into the same point at the same time. Due to the heterogeneity of images and point clouds, the two types of features are not in the same feature space. To address this problem, the proposed MFA module can effectively fuse the two types of features to get fused features. Subsequently, the fusion features guided by keypoints are processed by the region proposal network (RPN) module to get ROIs. Finally, ROIs are further reduced by ROI pooling, and the location and class information of the object is obtained by the refinement network.

### 3.1. P-FPS

Many point-based LiDAR 3D object detection algorithms [51,52] downsample the raw point cloud by FPS to get the keypoints to reduce the computational effort. Projection-based fusion algorithms [20,53] will project the LiDAR point cloud onto the image and often use this downsampling method to process the raw point cloud. However, FPS uniformly samples point clouds in 3D space, which results in uneven distribution of point clouds when projected onto an image. As shown in Figure 2, distant objects contain little feature information because they contain relatively few pixels. In contrast, the close object contains more pixels and can provide rich features. The key points obtained by FPS are mostly distributed in the distance. The image features sampled in this way lose considerable detail.

To solve the above problems, a P-FPS is proposed for point cloud sampling. As shown in Figure 3, suppose there are *M* 3D spatial coordinates $P_{Li}\left(x_{i},y_{i},z_{i}\right)\in R^{L},i=1,2\cdots M$ in the raw point cloud, which can be projected onto the image plane by Equation (2) to obtain the corresponding 2D pixel coordinates $P_{Ii}\left(u_{i},v_{i}\right)\in R^{I},i=1,2\cdots M$. In 3D space, the FPS requires the calculation of the 3D Euclidean distance between the coordinates of two points, $P_{Li}$ and $P_{Lj}$. Correspondingly, in the image plane, the 2D Euclidean distance $D_{I}\;$between the pixel coordinates $P_{Ii}$ and $P_{Ij}$ is required to be calculated, which can be expressed as:(3)$$D_{I}=\|F_{p}\left(x_{i},y_{i},z_{i}\right)-F_{p}{\left(x_{j},y_{j},z_{j}\right)\|}_{2},i,j=1,2\cdots M$$
where $F_{p}\left(\cdot \right)$ is the projection transformation relationship from the point cloud to the image plane, which can be obtained from Equation (2). After the image plane is processed by 2D FPS, the close object can contain more points. With the index information of these points, the keypoints in 3D space can be obtained. The specific content of the algorithm is shown in Algorithm 1.

The sampling method above allows closer points to be obtained, but results in a slight reduction of distant points. To balance this loss, the number of points obtained by FPS and 2D-FPS is divided by the following equation:(4)$$Num_{P-\mathrm{FPS}}=[\alpha \cdot Num_{2\mathrm{DFPS}}+\left(1-\alpha \right)\cdot Num_{\mathrm{FPS}}]$$
**Algorithm 1** 2D FPS$\mathrm{Input}:\;\mathrm{Raw}\;\mathrm{point}\;\mathrm{set}\;P_{Li}\in R^{L},\;i=1,2\cdots M$$\mathrm{Output}:\;\mathrm{Keypoints}\;\mathrm{set}\;P_{Kj}\in R^{L},\;j=1,2\cdots T$1 **Suppose**
*t* keypoints has been sampled and (t + 1)th keypoint is sampled as follows2 **for**
i=0
**to**
$\left(M-t\right)$ by 1 **do**3  **for**
j=0
**to**
t by 1 **do**4   Calculate the distance between point $P_{Li}$ and point $P_{Kj}$: $D_{I}=\|F_{p}\left(P_{Li}\right)-F_{p}{\left(P_{Kj}\right)\|}_{2}$5   **if**
$D_{I}<$ last minimum dist **then**6    dist = $D_{I}$7   **end if**8  **end for**9 **end for**10 Select the point with minimum dist as the keypoint11 **return** Keypoints set

### 3.2. Point-Guide Feature Abstract

**3D Feature Abstraction.** The raw point cloud is first divided into many voxels of size d × w × h. The feature vector of each non-empty voxel is encoded as:(5)$$V_{i}=\left\{\frac{\sum x_{i}}{Point\_num},\frac{\sum y_{i}}{Point\_num},\frac{\sum z_{i}}{Point\_num},\frac{\sum r_{i}}{Point\_num}\right\}$$
where *Point_num* is the number of points contained within each voxel and $r_{i}$ is the reflection intensity of that point. Then, the 3D sparse convolution of 3 × 3 × 3 is used to obtain spatial features at four different downsampling scales (1×, 2×, 4×, 8×).

**2D Feature Abstraction.** Image features contain rich texture and color information. In order to be consistent with 3D feature extraction, ResNet-18 [54] is utilized to extract image features. The network uses a series of 3 × 3 2D convolutional kernels to downsample the feature maps at four different scales. The four scales of feature maps focus on different feature details, and the same downsample scale as the 3D backbone is adopted. 

**Point Feature Sample.** After obtaining T keypoints $P_{K}=\left\{p_{K1},\cdots ,p_{Kj}\right\},\;j=1,2\cdots T$ by P-FPS, each point will aggregate 3D and 2D features, respectively. For 3D features, each scale feature is first sampled using ball query. Suppose that the kth layer Voxel-wise features are expressed as:(6)$$V^{k}=\left\{v_{1}^{k},\cdots ,v_{N}^{k}\right\}$$

The 3D coordinates of the kth layer are expressed as:(7)$$V^{c}=\left\{v_{1}^{c},\cdots ,v_{N}^{c}\right\}$$

The feature vectors of all voxels within the radius *r* are encoded as:(8)$$F_{i}^{r}=\left\{{\left[v_{i}^{k};v_{i}^{c}-p_{Ki}\right]}^{T}|\begin{array}{c} \|v_{i}^{c}-p_{Ki}{\|}^{2}<r\\ \forall v_{i}^{c}\in V^{c},\forall v_{i}^{k}\in V^{k} \end{array}\right\}$$
where $v_{i}^{c}-p_{Ki}$ denotes the local space features. Subsequently, these voxel features are aggregated to the keypoint $p_{Ki}$ as follows:(9)$$F_{i}^{3Dk}=max\left\{MLP\left(G\left(F_{i}^{r}\right)\right)\right\}$$
where $G\left(\cdot \right)$ denotes the random sampling of t voxels features. The feature vectors of all 4 scales are concatenated to the keypoint $p_{Ki}$ as follows:(10)$$F_{i}^{3D}=\left[F_{i}^{3D1},F_{i}^{3D2},F_{i}^{3D3},F_{i}^{3D4}\right]$$

For 2D features, the keypoints cannot be accurately corresponded to individual pixel points, although they have been projected onto the pixel plane. The reason is that the projection point coordinates are continuous, while the pixel point coordinates are discrete. Therefore, bilinear interpolation is employed to sample the four pixels around the projection point, and this process can be expressed as:(11)$$F_{i}^{2Dk}=\left\{B\left(F_{i1}^{Ik},F_{i2}^{Ik},F_{i3}^{Ik},F_{i4}^{Ik}\right)\right\}$$
where *B* denotes the bilinear interpolation function and $F_{i1}^{Ik},F_{i2}^{Ik},F_{i3}^{Ik},F_{i4}^{Ik}$ denote the feature vectors of the four neighboring pixel points around the projection point. Finally, as with the 3D features, the 2D features are stitched to the same keypoint $p_{Ki}$
(12)$$F_{i}^{2D}=\left[F_{i}^{2D1},F_{i}^{2D2},F_{i}^{2D3},F_{i}^{2D4}\right]$$

### 3.3. MFA Fusion

Since DETR [40] introduced the self-attention transformer [39] into the field of object detection, many studies [36,41,42,45] have attempted to use the attention mechanism for 2D or 3D object detection. Most of these approaches focus on spatial attention, and in this paper, we propose an MFA module based on the attention of feature channels. The purpose of this module is to allow point cloud features to adaptively fuse image features based on feature correlations. This module improves the robustness of the detection algorithm when the image or point cloud contain noise.

As shown in Figure 4, for a keypoint $p_{Ki}$, the image features $F_{i}^{2Di}$ at the *k*th layer scale are encoded by two sets of *N* 1D convolution kernels to obtain *N* Image value $V_{in}^{k}$ and Image key $K_{in}^{k}$, respectively, while the point cloud features $F_{in}^{3Dk}$ are encoded by 1D convolution kernels to obtain Point query $Q_{i}^{k}$. Each feature point is sampled to obtain a 1D feature vector, therefore a 1D convolution kernel is used to encode the image feature vector and the point cloud feature vector to obtain the same feature space. The vector length of each layer is set to [32,32,64,128] after the 1D convolution kernel encoding.

In the spatial self-attention mechanism of DETR [40], a positional encoding is added to the feature vector to enable the network to learn positional features. In the MFA module, the feature vectors used for attention calculation come from the same location, thus there is no need to add position encoding. Image key and Point query are used to calculate the feature weights. The correlation vector $RE_{}^{k}$ and feature weight $W^{k}$ are obtained by summation of the dot product, and the formula is as follows:(13)$$RE_{}^{k}=\left[\begin{array}{c} re_{1}^{k}\\ re_{2}^{k}\\ \cdots \\ re_{j}^{k} \end{array}\right]=\left[\begin{array}{c} SUM(Q_{i}^{k}\cdot K_{i1}^{k})\\ SUM(Q_{i}^{k}\cdot K_{i2}^{k})\\ \cdots \\ SUM(Q_{i}^{k}\cdot K_{ij}^{k}) \end{array}\right]\frac{1}{\sqrt{d_{k}}},j=1,\ldots ,N$$
(14)$$W^{k}=\mathrm{Softmax}\left(RE_{}^{k}\right)=\left[\begin{array}{c} w_{1}^{k}\\ w_{2}^{k}\\ \cdots \\ w_{j}^{k} \end{array}\right],j=1,\ldots ,N$$
where $d_{k}$ is the dimension of Image key and Point query, which is used as a scale factor to prevent extreme values after the summation of the dot product. Then the softmax function is used to score the *N* elements in the correlation vector $RE_{j}^{k}$ to get the feature weight $W^{k}$, which represents the correlation between different image features and point cloud features. Feature weight is weighted and summed with *N* Image values $V_{in}^{k}$ to obtain the final image feature vector, which is summed with Point query $Q_{i}^{k}$ to obtain the final fusion feature vector $F_{i}^{Fk}$.
(15)$$F_{i}^{Fk}=J(w_{j}^{k}\cdot V_{ij}^{k})+Q_{i}^{k}$$
where *J*(∙) denotes the summation of the dot product.

The raw point-cloud feature is aggregated as in Equation (8). For the BEV, we project the keypoint to the 2D bird-view coordinate system, and utilize bilinear interpolation to obtain the features from the bird-view feature. Finally, to enrich the features, each keypoint-guided feature vector is joined with feature vectors representing the raw point cloud and BEV. It can be expressed as:(16)$$F_{i}^{k}=\left[F_{i}^{F1};F_{i}^{F2};F_{i}^{F3};F_{i}^{F4};F_{i}^{raw};F_{i}^{BEV}\right]$$

After downsampling by P-FPS, although more points fall within the object, there are still many background points that do not contribute much to the detection task. Therefore, following the design of PV-RCNN, the predicted keypoint weighting (PKW) module is used to predict the probability of each keypoint being a foreground point. The difference is that the keypoint features processed by PKW contain information from images and point clouds. The process of predicting keypoints can be expressed as:(17)$${\tilde{F}}_{i}^{p}=A\left(F_{i}^{k}\right)\cdot F_{i}^{k}$$
where $A\left(\cdot \right)$ denotes a three-layer MLP network with a sigmoid function to predict foreground confidence between [0, 1]. 

### 3.4. ROI Pooling and Detection Head

The RPN structure is the same as PV-RCNN, where many ROIs are predicted based on 3D spatial features. As shown in Figure 5, the keypoints feature vectors contained within each ROI are aggregated to a fixed 6 × 6 × 6 grid point in order to obtain a more accurate prediction of the bounding boxes. The process is similar to the point feature abstraction process, where the keypoints around the grid points are sampled by using ball queries at different scales. The final ROI contains 216 grid point feature vectors within each ROI, and then the class and 3D bounding boxes of the object are predicted by two layers of MLP. 

## 4. Experiments

### 4.1. Datasets

KITTI [55] is a common benchmark for outdoor 3D object detection. There are 7481 training samples and 7518 test samples, where the training samples are generally divided into the train split (3712 samples) and the val split (3769 samples). In the 3D object detection task, Car, Pedestrian and Cyclist were evaluated separately. KITTI presents three object detection tasks (Easy, Moderate and Hard) of different difficulty according to the size of the object, the degree of occlusion and the degree of truncation. In the experimental results, we adopted “Easy”, “Mod.” and “Hard” to represent these three tasks.

Most of the KITTI data are collected in good weather, however there are many real scenarios with bad weather (e.g., rain) that can affect the accuracy of LiDAR and camera outputs. To simulate the rainy scenarios, two noises are added separately. The camera is often influenced by rain which causes blurring and occlusion of the output image, resulting in the loss of detailed features of the object [7]. In this paper, this blurring is simulated using Gaussian blurring as follows:(18)$$\left\{\begin{array}{c} F\left(u\right)=\frac{1}{\sqrt{2\pi \sigma }}e^{-\frac{u^{2}}{2\sigma_{u}^{2}}}\\ F\left(v\right)=\frac{1}{\sqrt{2\pi \sigma }}e^{-\frac{v^{2}}{2\sigma_{v}^{2}}} \end{array}\right.$$
where *u*, *v* denote the local coordinates of the Gaussian kernel and $\sigma_{u}^{2},\sigma_{v}^{2}$ denote the variance in both directions. The Gaussian kernel is a two-dimensional normally distributed matrix containing a set of pixel weights that can smooth the pixels inside the Gaussian kernel. For the occlusion effect of raindrops, we randomly generated different raindrop traces through image processing. 

LiDAR is prone to noise in rainy environments [56,57], because raindrops can cause multiple reflections to LiDAR. The result of this is that the 3D contour boundary of the object is less obvious. Random noise is added to the 3D coordinates of the raw point cloud to simulate this effect. 

Figure 6 shows the effect after adding the noise. It can be observed that the vehicle textures in the images become blurred and the vehicle contours in the point cloud are no longer regular. These changes increase the difficulty of object detection because the object specific features are weakened.

### 4.2. Training and Inference Details

Since the camera FOV is in front of the vehicle, this paper takes the camera FOV as the boundary and removes the point clouds beyond the boundary. Meanwhile, a rectangular area is set to filter the remaining point clouds. The range of this region is (0 m < x < 70.4 m, −40 m < y < 40 m, −1 m < z < 3 m), which is divided into many voxels of size (0.05 m, 0.05 m, 0.1 m). 

For training, both the 2D backbone and 3D backbone use pretrained parameters directly, and the parameters are not updated. In addition, since there is a fixed transformation relationship between LiDAR and camera, and the fusion data augmentation scheme is not mature, conventional data augmentation methods are not used in this paper. The network is built using OpenPCDet and trained on a GTX3090 with batchsize 4, epoch 80, and learning rate 0.01. 

### 4.3. Results on the KITTI Dataset with Rain Noise

The mainstream object detection algorithms are tested on the generated KITTI with rain noise. Table 1 shows the test results of each algorithm in a KITTI validation split with rain noise. For better evaluation, all algorithms are retrained on data with rain noise, and the same training method is used. As can be seen in Table 1, the proposed algorithm shows the best results in all three detection tasks of Car, Pedestrian and Cyclist. Compared with the baseline algorithm PV-RCNN in this paper, the mAP is improved by 5.86%, 21.15% and 41.17% in the three categories of the 3D detection task, respectively. Meanwhile, Table 2 demonstrated that the proposed algorithm is still the optimal result in the BEV task. Compared with PV-RCNN, the mAPs of the three categories are improved by 3.76%, 16.80% and 40.00%, respectively. In the detection tasks of Pedestrian and Cyclist, the algorithm in this paper shows a relatively large lead. The reason is that Pedestrian and Cyclist contain few point clouds and their features are more affected by rain noise. It causes difficulties in point cloud-based algorithms in both training and detection, and the proposed algorithm can aggregate image features as a complement to enrich the overall feature information.

Figure 7 shows the comparison of PR curves for different detection algorithms on KITTI with rain noise. Recall is the proportion of the number of objects that are correctly detected among all ground truths. The precision in the Figure 7 indicates the percentage of detected objects that are correctly detected at different recall rates. From the figure, it can be observed that the proposed algorithm achieves the highest accuracy rate at the recall rate of [1–0.6]. The high accuracy is also maintained at the recall rate of [0.6–0]. 

In order to compare the robustness of the algorithms to the effects of rain noise, pre-trained models are directly used for testing on noisy datasets. The proposed algorithm is trained on the raw KITTI without rain noise. As can be seen in Table 3, the proposed algorithm still achieves the best results even without training on noisy data. 

### 4.4. Results on the NUSCENES Dataset

To validate the effectiveness of the proposed algorithm, it was evaluated and compared on the NUSCENES dataset. The dataset uses six cameras, one 32-line LiDAR, five millimeter wave radars, IMU and GPS with 1000 different urban scenes, including scenes with visual impact such as night and rain. As shown in Table 4, the proposed algorithm in the NUSCENES dataset achieves a competitive performance. The proposed algorithm achieves the highest results in the mAP metric. The rainy and night scenes contained in the dataset affect the camera more, and the proposed algorithm adaptively learns the association of image features with point cloud features in different feature spaces, thus achieving a high robustness.

### 4.5. Results on Physical Test Platform

As shown in Figure 8, a physical test platform was built to verify the algorithm proposed in this paper. The platform is installed with a 32-line LiDAR, an IMU, a monocular camera, and a 77 GHz millimeter wave radar. Data from all sensors were collected by VECTOR VN5640 and aligned to the same timestamp with same time source. The proposed algorithm was tested on LiDAR and camera data, which were converted to the format of the KITTI dataset.

As shown in Figure 9, a slice of data collected by the actual physical test platform shows that the proposed algorithm obtains better 3D detection performance. Our algorithm and PV-RCNN are trained on KITTI data with noise, while noise is added on the actual test data. The results in the first row are output by the camera, the results in the second row are obtained by the proposed algorithm, and the results in the third row are obtained by the PV-RCNN. It can be observed that the proposed algorithm is able to detect more objects. It is important to note that testing the algorithm on the collected data is a cross-domain problem. This is due to the fact that the KITTI sensor models and scenarios are not the same as the collected data, and random noise simulating the effects of weather is added during the testing. The proposed algorithm incorporates information from different stage features and different feature spaces to effectively alleviate this problem. 

## 5. Ablation Studies

### 5.1. On P-FPS

To study whether the points sampled by the proposed P-FPS contribute to the object detection task, the recall of ground truth points is used to evaluate the results. The object bounding box provided by the ground truth is used to determine whether the points sampled by P-FPS and FPS belong to the foreground points. As shown in Figure 10, the number of sampled points falling into the ground truth object box by P-FPS is approximately three times higher than that by FPS for the same setting of keypoints number. From the results, it is clear that the proposed P-FPS can sample more points belonging to the close objects, which is helpful for both the effectiveness of image feature sampling and the subsequent object bounding boxed prediction stage.

While increasing the number of close object points, the distant points should not be ignored. Therefore, different proportions are assigned for the sampling points number of FPS and 2D-FPS by Formula (4). As shown in Figure 11, the effects of different α are compared on the detection performance, and three categories of mAP are used to evaluate the detection results. It can be seen that the proposed method achieves the optimal detection performance at α = 0.3. When α decreases, the sampling results tend to be FPS, and the keypoints obtained do not contain closer object points. On the contrary, when α increases, the sampling results tend to be P-FPS, and the keypoints lose the features of distant objects. Table 5 shows the 3D detection results of the proposed sampling method compared with the traditional FPS method. The proposed sampling algorithm is substantially better than FPS in Car and Pedestrian, and obtains competitive results in Cyclist. 

### 5.2. On Fusion Feature

PKW module can predict the probability of keypoints being foreground points, and the accuracy of this result will directly affect the subsequent 3D bounding box prediction. Keypoints with predicted probability greater than 0.05 are marked as true positives. It can be observed from Figure 12a,b that by fusing the feature information of the image, the foreground points predicted by the PKW module are more accurate. Conversely, if only LiDAR point cloud features are used, the PKW module will predict more background points. The results confirm that the PKW module will assign more weights to the foreground points by using the fused features, which further reduces the influence of background points.

The detection performance of fusing features at different stages is evaluated separately. As shown in Table 6, the first 4 rows are the result of using only one of the features in $\left\{F_{i}^{F1},F_{i}^{F2},F_{i}^{F3},F_{i}^{F4}\right\}$. It can be observed that the $F_{i}^{F1},F_{i}^{F2}$ features obtain higher detection accuracy, which indicates that low-dimensional features contribute more to the detection algorithm. The last row adopts the fused features of all stages and obtains the best detection accuracy.

### 5.3. On MFA

In the attention mechanism, multiple heads are usually used to fuse representations from different feature spaces. The detection performances of the proposed algorithm with different numbers of heads are evaluated. As shown in Figure 13, in the Car category, the best detection performance is achieved when the number of heads is 4. When the number of heads decreases or increases, the detection accuracy decreases.

The image features contain rich texture features. In order to verify the effectiveness of these features in improving the detection performance in rain scenarios, the image features are removed and compared with the proposed algorithm. It can be seen from Table 7 that only using point cloud features will significantly reduce the detection performance of the algorithm for car category. This suggests that even if the image features become blurred, there is still a non-negligible contribution to the detection accuracy. 

To visually demonstrate the attention effect of MFA, the attention results of each keypoint are visualized in Figure 14. The first row in the figure is the raw image output by the camera, the second row is the attention difference of each feature point, and the third row is the detection result. Each keypoint aggregates image features in four different feature spaces through the MFA module, and these features are assigned different weights by the attention mechanism. The different colors shown in the Figure 14 represent the image features with the largest weight among the four kinds of image features fused by the keypoint. It can be observed that the proposed method adaptively selects different image feature fusion weights at different locations in the image. There is a clear boundary difference between most objects and backgrounds. This shows that the MFA module can selectively fuse image features according to the illumination changes and texture details of the image.

## 6. Conclusions

In this paper, a new camera and LiDAR-based 3D object detection algorithm is proposed, which is based on point-guided sampling of point cloud features and image features. The multi-head attention mechanism is introduced into the feature fusion task, and the proposed MFA mechanism is used to realize the adaptive fusion of point cloud features and image features. A projection-based sampling method, P-FPS, is proposed, which can significantly improve the sampling rate of image features and obtains more abundant image information to the fused features. In some scenes, the proposed F-FPS can extract about three times the number of effective feature points, compared with traditional FPS. In order to simulate the scene where the sensor contain rain noise, the KITTI dataset with rain noise is established. The proposed method shows better detection performance and robustness when tested on KITTI and NUSCENES datasets. Compared with the baseline algorithm PV-RCNN in this paper, the mAP is improved by 5.86%, 21.15% and 41.17% in the three categories of the 3D detection task, respectively. Finally, the algorithm is tested on the physical test platform, which further verifies the effectiveness of the algorithm. In future work, more real extreme weather data will be collected, and weather simulation tests in virtual environments will be used to test the robustness of the algorithm. In addition, the domain adaptation problem caused by the different distributions of the training dataset and the testing dataset will also be studied to improve the generalization of the feature extraction network.

## Figures and Tables

**Figure 1 sensors-23-00233-f001:**
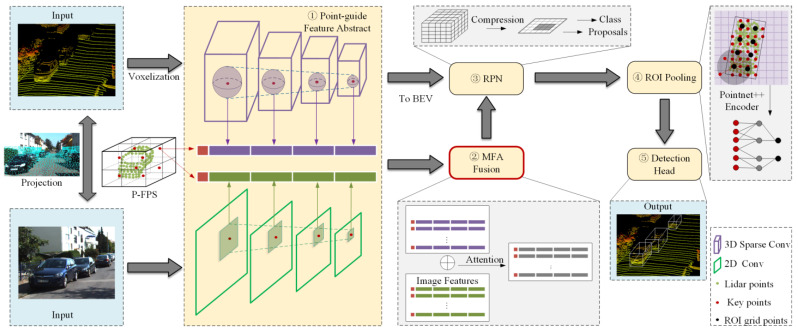
The architecture of the proposed 3D object detection.

**Figure 2 sensors-23-00233-f002:**
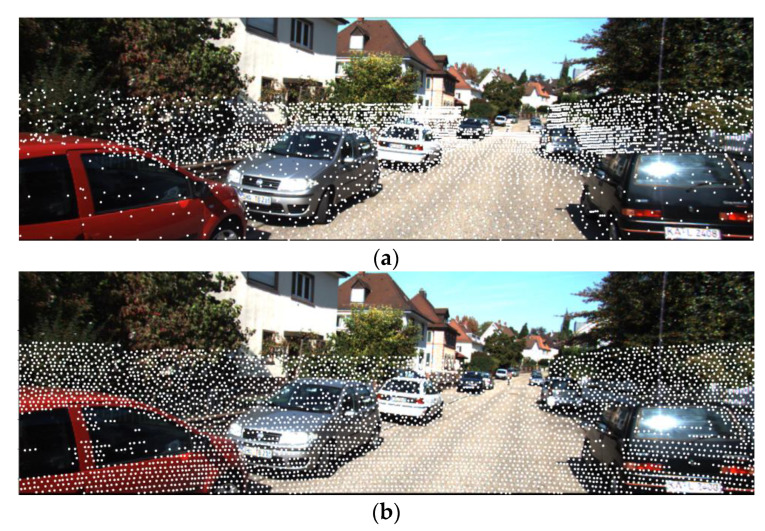
Comparison of (**a**) FPS and (**b**) P-FPS.

**Figure 3 sensors-23-00233-f003:**
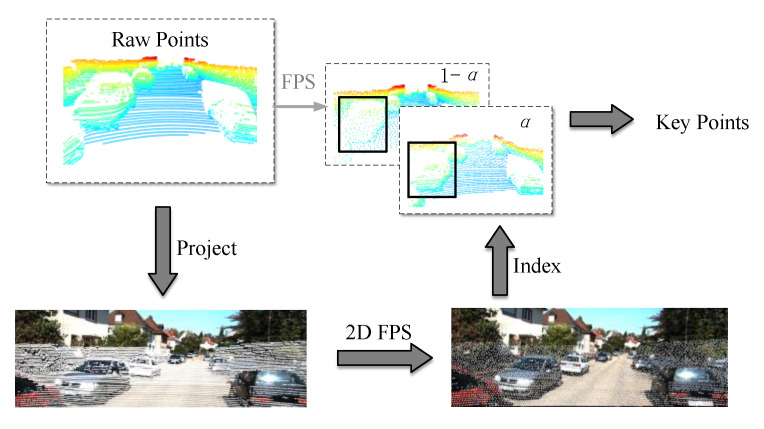
Illustration of the P-FPS.

**Figure 4 sensors-23-00233-f004:**
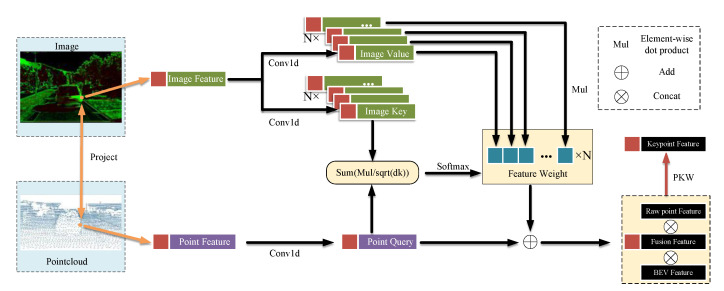
Multimodal Feature Attention (MFA) fusion module.

**Figure 5 sensors-23-00233-f005:**
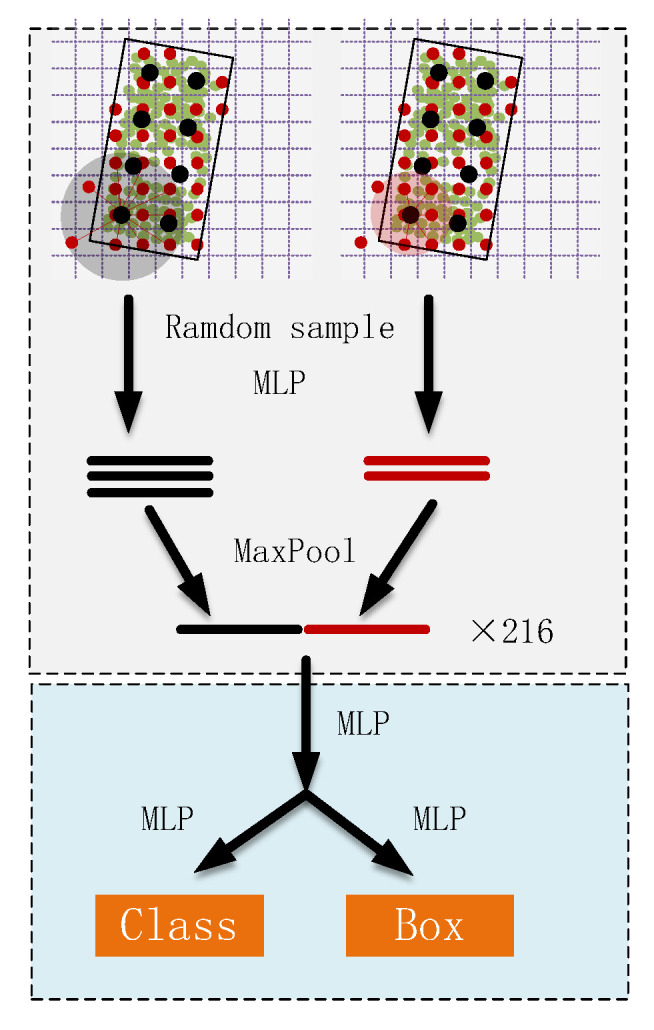
Region of Interest (ROI) pooling module and detection head.

**Figure 6 sensors-23-00233-f006:**
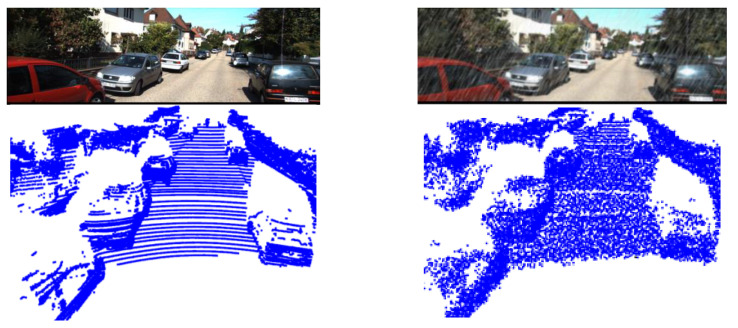
The effect of adding rain noise to the data.

**Figure 7 sensors-23-00233-f007:**
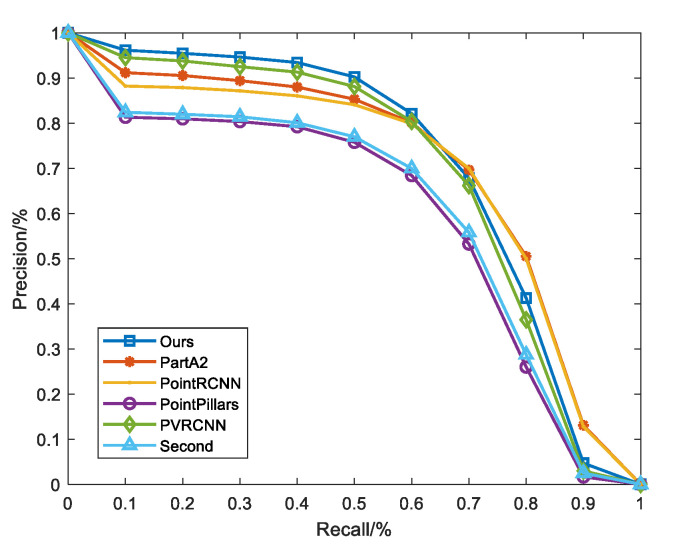
Precision–recall curve.

**Figure 8 sensors-23-00233-f008:**
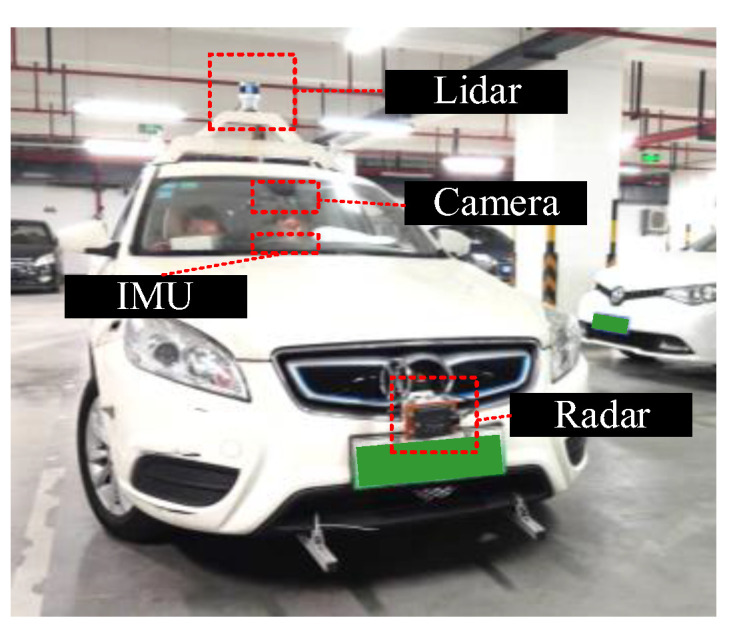
Physical test platform.

**Figure 9 sensors-23-00233-f009:**
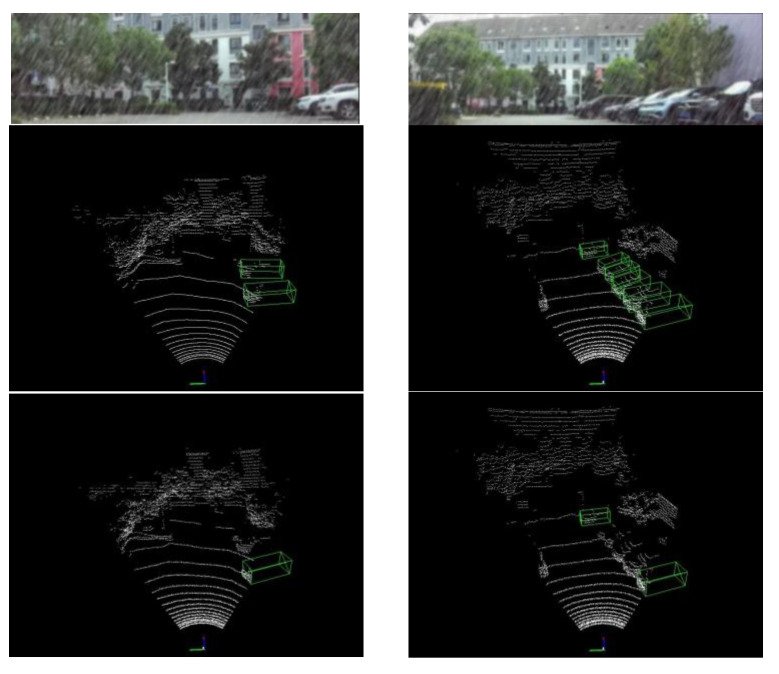
Visual results of physical test platform dataset.

**Figure 10 sensors-23-00233-f010:**
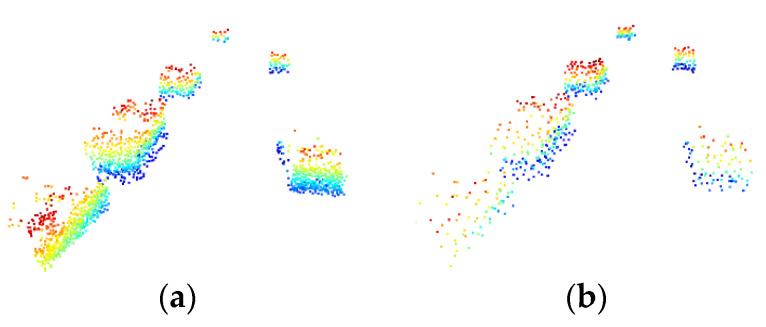
Comparison of P-FPS and FPS on recalling GT points: (**a**) P-FPS (recall 1507 points); (**b**) FPS (recall 575 points).

**Figure 11 sensors-23-00233-f011:**
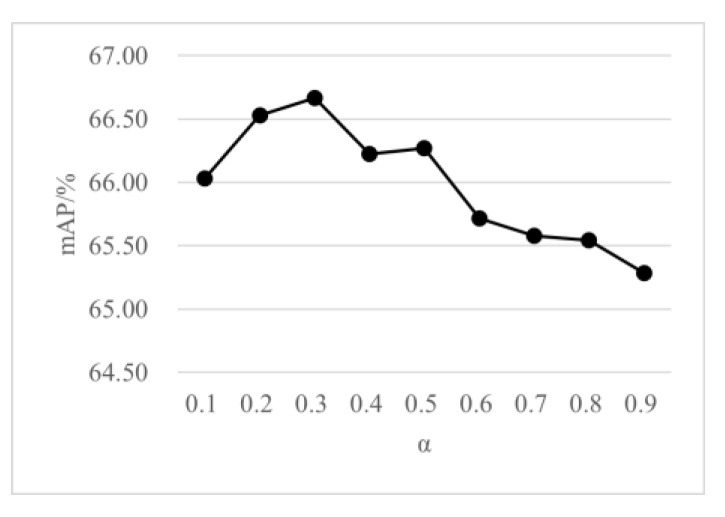
Performance comparisons on different α.

**Figure 12 sensors-23-00233-f012:**
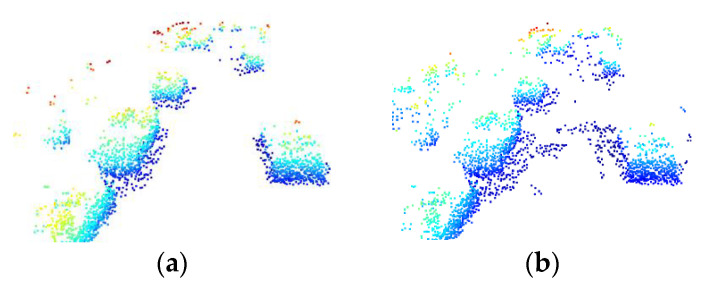
Effect of fusion feature on Predicted Keypoint Weighting (PKW). (**a**) Foreground point prediction with fusion feature. (**b**) Foreground point prediction without fusion feature.

**Figure 13 sensors-23-00233-f013:**
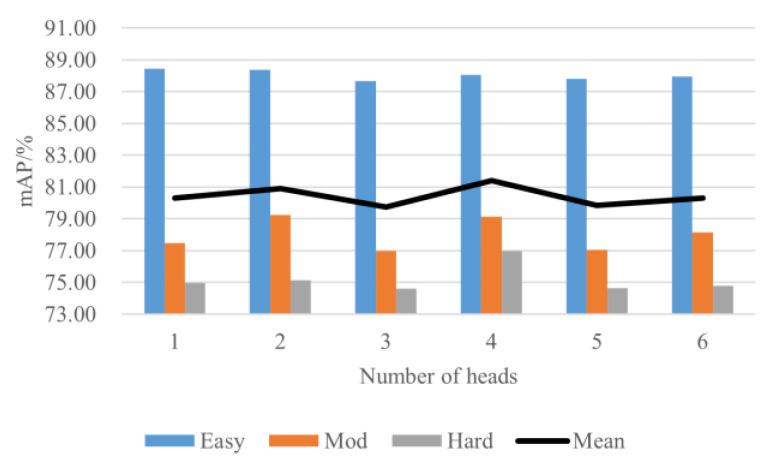
Performance comparison on the car category with different heads.

**Figure 14 sensors-23-00233-f014:**
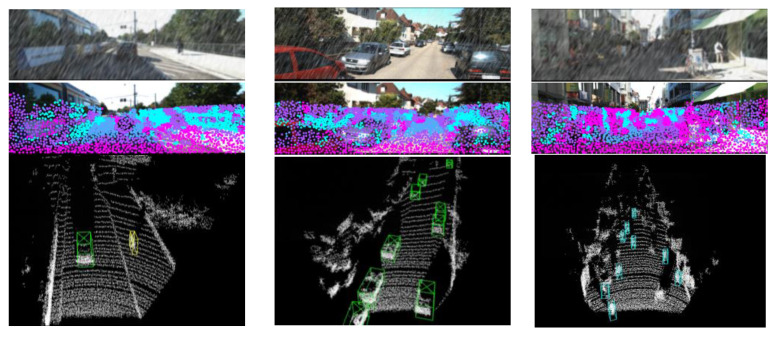
Visualization of MFA. Green/blue/yellow boxes indicate Car/Pedestrian/Cyclist.

**Table 1 sensors-23-00233-t001:** Results of 3D detection on the KITTI validation split with rain noise after training.

	Modality	Car				Pedestrian				Cyclist			
		Easy	Mod.	Hard	mAp	Easy	Mod.	Hard	mAp	Easy	Mod.	Hard	mAp
PV-RCNN [1]	L	85.32	74.28	69.96	76.52	46.71	41.71	38.52	42.31	61.23	40.13	36.51	45.96
PointRCNN [4]		85.59	76.74	72.58	78.30	47.90	40.54	34.47	40.97	80.44	51.49	49.15	60.36
PartA2		88.91	77.31	74.57	80.26	56.79	49.38	43.17	49.78	76.31	51.72	47.44	58.49
SECOND [2]		80.24	67.93	64.45	70.87	41.82	35.37	31.68	36.29	61.32	42.46	40.43	48.07
PointPillars [5]		74.58	61.73	59.70	65.34	32.79	28.00	24.51	28.43	55.66	35.45	33.41	41.51
EPNet [20]	L + C	87.96	76.53	75.24	79.91	-	-	-	-	-	-	-	-
MFA fusion (Ours)		88.24	77.95	76.85	81.01	56.82	50.49	46.47	51.26	84.46	60.20	57.59	64.88

**Table 2 sensors-23-00233-t002:** Results of Bird’s Eye View (BEV) on the KITTI validation split with rain noise after training.

	Modality	Car				Pedestrian				Cyclist			
		Easy	Mod.	Hard	mAp	Easy	Mod.	Hard	mAp	Easy	Mod.	Hard	mAp
PV-RCNN [1]	L	92.12	83.95	79.39	85.15	52.93	47.45	44.21	48.20	62.70	41.15	39.00	47.61
PointRCNN [4]		91.95	84.26	81.97	86.06	54.37	44.82	38.44	45.88	81.09	53.97	49.54	61.53
PartA2		92.87	84.13	83.63	86.88	63.44	55.51	48.96	55.97	77.37	54.59	50.23	60.73
SECOND [2]		89.93	82.01	79.95	83.96	51.59	44.25	40.22	45.35	64.20	45.96	43.78	51.31
PointPillars [5]		89.76	79.51	77.30	82.19	40.67	34.10	31.53	35.43	58.23	37.61	35.59	43.81
EPNet [20]	L + C	92.16	85.89	84.27	87.44	-	-	-	-	-	-	-	-
MFA fusion (Ours)		92.49	86.31	86.24	88.35	61.85	55.62	51.43	56.30	81.19	60.74	58.02	66.65

**Table 3 sensors-23-00233-t003:** Results of 3D detection on the KITTI val split with rain noise before training for the Car category.

	Modality	Car			
		Easy	Mod.	Hard	mAp
PV-RCNN [1]	L	34.90	31.00	29.07	31.66
PointRCNN [4]		27.04	25.61	23.54	25.40
PartA2		20.02	21.64	20.24	20.63
SECOND [2]		36.76	33.30	31.91	33.99
PointPillars [5]		30.82	27.87	27.26	28.65
MFA fusion (Ours)	L + C	37.07	33.73	31.43	34.08

**Table 4 sensors-23-00233-t004:** Results of detection on the NUSCENES dataset.

	mATE	mASE	mAOE	mAVE	mAAE	mAP	NDS
PointPillars [5]	33.87	26.00	32.07	28.74	20.15	44.63	58.23
SECOND [2]	31.15	25.51	26.64	26.26	20.46	50.59	62.29
CenterPoint [58]	30.11	25.55	38.28	21.94	18.87	56.03	64.54
MFA fusion (Ours)	35.56	28.33	35.29	29.75	22.46	57.54	60.51

**Table 5 sensors-23-00233-t005:** Results of 3D detection on the rain noise data with different sample methods.

	Car				Pedestrian				Cyclist			
	Easy	Mod	Hard	mAp	Easy	Mod	Hard	mAp	Easy	Mod	Hard	mAp
FPS	88.14	76.58	76.23	80.32	56.15	50.32	44.82	50.43	83.61	61.89	57.38	67.63
P-FPS	88.24	77.95	76.85	81.01	56.82	50.49	46.47	51.26	84.46	60.20	57.59	67.42

**Table 6 sensors-23-00233-t006:** Effects of different feature components.

$F_{i}^{F1}$	$F_{i}^{F2}$	$F_{i}^{F3}$	$F_{i}^{F4}$	mAP (Car)
√				80.54
	√			79.91
		√		79.25
			√	77.36
√	√	√	√	81.01

**Table 7 sensors-23-00233-t007:** The suppression effect of fusion features under rain noise.

Feature	Easy	Mod	Hard	mAP (Car)
Only LiDAR	88.05	74.21	70.49	77.58
MFA fusion	88.24	77.95	76.85	81.01

## Data Availability

Publicly available datasets were analyzed in this study. These data can be found here: https://www.cvlibs.net/datasets/kitti/ and https://www.nuscenes.org/nuscenes.

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
