# Peer review of "Anti-Noise 3D Object Detection of Multimodal Feature Attention Fusion Based on PV-RCNN"

_sensors, 2022, doi:10.3390/s23010233_

Round 1

Reviewer 1 Report

This paper proposes a point-guided Multimodal Feature Attention (MFA) fusion 3D object detection algorithm to solve the mismatch of physical characteristics of camera and LiDAR and improve the robustness of 3D object detector against noise. Although the structure of this manuscript is clear and the problems to be solved are interesting, there are still several critical issues to be improved. Therefore, I suggest the authors conduct a major revision on this version:

For your convenience, the review comments are enumerate as follows:

1.     First and foremost, it is necessary to clarify the types and characteristics of noise, and how these noises affect the performance of image information, point cloud information, and 3D object detector. Please add these analysis of above issues in the Introduction.

2.     Please add the ablation studies on fusing two kinds of information to suppress the noise in 3D object detection.

3.     The attention-based method introduced in DETR is self-attention module. Please note the accuracy of the relevant references and descriptions.

4.     Please check whether the full name is provided when the abbreviation first appears.

5.     In section 3.1, please check the expression of 2D pixel coordinates. 

6.     In section 4.1, the mainly problem is robustness of 3D object detection under noisy conditions. However, the current manuscript only use simple blurring to simulate the actual noise. This experimental setup is not convincing, so please add verification experiments with more complex noise (such as the point-type noise simulating rainwater.).

Reviewer 2 Report

1. Please describe the variables in the formula clearly. For example, what does BEV represent in Formula 15? Please explain clearly how to obtain BEV feature vector.

2. Whether the characteristic dimensions of Formula 15 are the same, and explain what standard is used to implement concat?

3. Please explain what Easy, Mod. and Hard represent in the table 1? According to the experimental results, in the case of Easy, the results of proposed method are not the best. Please explain why.

4. There is less comparative analysis on the effect of the attention weight. It is suggested that comparative experiments on the effect of weight adjustment should be added.

Reviewer 3 Report

1. Please describe more clearly and specifically for your methods and contributions of this research in the abstract part.

2. The conclusion section would be improved with more quantitative data from the results. Also, according to the topic of the paper, the authors may propose some interesting problem as future work in conclusion.

3. In section 3, the point cloud and image plane are described in line 160 and line 161. Please describe more clearly about the relationships of the point cloud and image plane. Also, show them in Figure 1.

4. What are parameters for your 3D detection system about Z?, K, and M in this paper? Please show parameters in section 3.1 P-FPS.

5. Table 5 shows the 3D detection results of the proposed sampling method compared with the traditional FPS method based on noise data. What is noise data in table 5? Are rainy pictures you used in this experiments?    

6. Please describe clearly in the paragraph for Table 1, 2, 3, 5, 7 about Easy, Mod. and Hard.

Round 2

Reviewer 1 Report

I have no comments.